# Contents of Polyamines and Biogenic Amines in Canned Pet (Dogs and Cats) Food on the Austrian Market

**DOI:** 10.3390/foods10102365

**Published:** 2021-10-05

**Authors:** Peter Paulsen, Susanne Bauer, Friedrich Bauer, Zuzana Dicakova

**Affiliations:** 1Institute of Food Safety, Food Technology and Veterinary Public Health, Unit of Food Hygiene and Technology, University of Veterinary Medicine Vienna, Veterinärplatz 1, 1210 Vienna, Austria; susanne.bauer@vetmeduni.ac.at; 2Department of Food Hygiene, Technology and Safety, University of Veterinary Medicine and Pharmacy in Košice, Komenskeho 73, 04181 Košice, Slovakia; Zuzana.Dicakova@uvlf.sk

**Keywords:** canned pet food, amines, fish, biogenic amine index

## Abstract

Biogenic amines accumulate in proteinaceous raw materials used for pet food production. In canned, sterilized food, amine levels of the ingredients are preserved and may both be indicative of hygiene deficiencies in the ingredients as well as for potential adverse effects to the animals feeding on it. We determined the contents of biogenic amines and polyamines (dansyl derivatives, high performance liquid chromatography) in a variety of canned food for dogs (n = 72) and cats (n = 114) on the Austrian market and compared the results with common quality indices. Contents of putrescine, cadaverine, and tyramine were below the limit of detection in >70% of samples (maximum values: 21.5, 98.4 and 32.5 mg/kg wet weight, respectively). Median contents of histamine, spermidine, and spermine were 14.5, 12.7, and 29.4 mg/kg, and maximum values were 61.6, 28.2, and 53.6 mg/kg wet weight, respectively. The sum of (putrescine + cadaverine + histamine + tyramine) was >50 mg/kg in 22.6% of samples. The biogenic amine index exceeded “1” in 26.7% of samples. Whilst cat food contained significantly higher amounts of tyramine, dog food contained significantly higher amounts of histamine and spermine. In canned cat food, the ingredient “fish” was identified as a statistically significant risk factor for a biogenic amine index > 1 (relative risk = 3.0 (95% confidence interval: 1.8–5.5)) and for (putrescine + cadaverine + histamine + tyramine) exceeding 50 mg/kg (relative risk = 2.4 (95% confidence interval: 1.2–4.6)), due to higher contents of cadaverine in food samples containing fish. While all samples met the limits suggested in pet food production, we could demonstrate that the inclusion of fish in the formulation bears a significant risk for higher cadaverine contents.

## 1. Introduction

Although commercial slaughter procedures and breeds of slaughter animals are designed to optimize yield of edible organs for human consumption, a substantial fraction of organs or parts thereof do not enter the food chain and have to be processed or disposed of in an economical and sustainable way [1]. Some are generally excluded from the food chain (e.g., cattle hides), whilst others are unfit for human consumption because of pathologies or alterations caused by the slaughter process itself (e.g., meat around the sticking wound). Finally, for commercial reasons, some offal produced in “food grade” quality is used for pet food manufacture [2]. In 2018, the annual turnover for pet foods (dog, cat, birds, reptiles, fish) in Austria amounted to EUR 597 million, and, for 2021, the average quantity of pet food bought by one household was estimated to be 24.3 kg (www.statista.com, accessed 12 September 2021).

In the European Union, basic safety of hygiene aspects of food and feed are regulated by the same legislation [3], and more detailed requirements for feed containing animal tissues are defined. In contrast to raw or dried pet foods, canned products undergo a heat treatment sufficient to kill vegetative bacteria and bacterial spores and thus guarantees that products in unopened vessels remain stable and safe for extended storage periods without the need for cold storage [4,5]. This ensures that, when raw materials with a higher risk of containing pathogenic bacteria [2] are used, they can still be processed into “safe” feed.

Reports of high numbers of spoilage bacteria on raw foods for dogs [6] and cats [7] indicate that such foods are either produced from already deteriorated material or have been stored inadequately [8,9]. Besides, pathogenic bacteria have been recovered in such feed [6,10,11,12] and, for *Salmonella*, it has been demonstrated that dogs may become infected and then shed the pathogen [13].

Arguably, canned, sterilized pet food is generally considered a safe alternative, with respect to biological hazards, such as bacteria, viruses, and parasites. However, pet food recalls have been reported due to contamination with mycotoxins, adulteration, or too high contents in vitamin D or too low supplementation of vitamin B [14]. Since companion animals often consume a uniform diet, with multiple food units from a single batch, their exposure to foodborne hazards can be high [14]. There is also concern about the quality of the raw materials processed into canned pet food. In proteinaceous raw materials, the degradation of proteins, either autolytic or caused by contaminant bacteria, can lead to the accumulation of biogenic amines. These amines generally withstand heating; thus, contents of biogenic amines in heat treated foods reflect the amine contents and autolytic or bacterial activity of the raw materials. Consequently, indices generated by either summing up contents of individual amines or by calculating ratios, have been proposed as spoilage indicators [9]. Amongst the protein-rich foods, fish has received much attention [15], either due to the abundance of free amino acids in the tissues in general, or an abundance of histidine in particular, which can be converted, by bacterial action, to histamine.

The significance of biogenic amines in foods is not limited to quality issues. Vasoactive amines (histamine, tyramine) can act on the cardiovascular system, although there are considerable variations in susceptibility [16]. However, “histamine intolerance” is not limited to humans, but is also reported in pet animal species [17,18,19]. Another issue is the content of polyamines (namely, spermidine and spermine, and its precursor, putrescine) and polycationic substances, which normally stabilize DNA and are abundant in metabolically active cells [20,21], as well as in tumour cells [22]. The usefulness of polyamine-free diets or the administration of polyamine-analoga to combat cancer has been explored; however, more recently, the positive effects of polyamines have gained particular interest [23] as an “anti-aging” substance [24].

For pet food, histamine has been identified as a relevant hazard [25], to be addressed in the quality assurance systems of the companies. The sum of all amines has been proposed as a freshness indicator [26]. When applying quality indicator values calculated for foods, it should be considered whether the matrix is comparable. The biogenic amine index (BAI) by Mietz and Karmas [27], for example, was developed for fish and does not consider tyramine, which is abundant in stored meat [28]. Moreover, the natural content of biogenic amines and polyamines in inner organs can be higher than in muscle tissue [29], and such inner organs are usually processed into pet foods.

Although data have been published on amine contents in feed components, such as meat and fish meal [30,31,32,33], there is scarce published data on amine contents in canned, sterilized foods for dogs and cats. In the pet food industry, there seems to be a consensus on the significance of these compounds [34], and appropriate analytical methods have been published [32,33,35]. In previous studies, we found higher contents of histamine and spermine in canned than in raw (bones-and-raw-food; BARF) dog food [7], and an indication that the use of fish as an ingredient results in higher amine contents [36].

The aim of our study was to give an overview on the contents of the biogenic amines putrescine, cadaverine, histamine, and tyramine, and of the polyamines spermine and spermidine in canned pet food for dogs and cats on the Austrian market, and to explore whether amine contents differed between food for dogs or cats, or by ingredients, in particular fish.

## 2. Materials and Methods

### 2.1. Samples

Within a period of 14 days in the year 2021, outlets of the five major supermarket chains in Vienna were visited and all available types of canned pet food (dog and cat) were identified; we considered different brands, composition, and vessel size. Feed for medical purposes or for special dietary requirements was not considered. In total, 72 samples of canned dog and 114 samples of cat food were obtained. We recorded the information on the label regarding the ingredients and on the moisture content.

### 2.2. Determination of Biogenic Amines

For determination of biogenic amines and polyamines, the content of the can was transferred in a plastic bag and manually kneaded (30 to 80 times depending on the volume of the can) and then a subsample of 100 g was mixed in a grinder. A 10 g portion was homogenized (Ultra-Turrax T25 blender; Jahnke and Kunkel, Staufen i. B., Germany) in 90 g 10% trichloroacetic acid. The slurry was filtered through a folded paper filter (MN 615 ¼; Macherey-Nagel, Düren, Germany) and through a 0.45 µm cellulose-acetate membrane (Roth, Germany). The filtrate was brought to alkaline pH (ca. 11) with NaHCO_3_ and reacted with dansylchloride at 70 °C 10 min as described previously [37]. The derivatives of the amines cadaverine, histamine, putrescine, tyramine, spermidine, and spermine were separated on an RP-C_18_-HPLC column (Symmetry, 4.6 × 150 mm; 3.5 µm; Waters, Milford, MA, USA) on a Waters 2695 station with a acetonitrile:methanol:acetic acid gradient elution program and quantified by UV-absorption (254 nm; Waters 996). Peaks were identified and quantified by the external standard method. Reagents were from Sigma–Aldrich (St. Louis, MO, USA) (amines and dansyslchloride) or Thermo Fisher (Waltham, MA, USA) (other reagents and eluents/solvents). Limit of detection was 1.5–2.8 mg/kg, depending on amine. Results were expressed in mg/kg fresh weight.

### 2.3. Index Values and Proposed Limits

For assessment of amine levels, we calculated the biogenic amine index [27] and an indicator value including tyramine contents [38].

Equation according to [27]:

Biogenic amine index, BAI = (putrescine + cadaverine + histamine)/(spermidine + spermine + 1) [27], with a value >1 indicative for hygienic deterioriation; amine contents in mg/kg fresh weight.

Index according to [38]:

Index = putrescine + cadaverine + histamine + tyramine, with values >50 indicative for deterioriation; amine contents in mg/kg fresh weight.

For pet foods in particular, the sum of all amines should not exceed 300 mg/kg [26], and histamine levels not 500 mg/kg [25].

### 2.4. Statistical Procedures

For statistical evaluation, we used non-parametric tests. Mann–Whitney test was used to compare the contents of histamine, spermidine, and spermine in cat vs. dog food. The chi-square test was applied to compare frequencies of tyramine results exceeding the limit of detection and the frequencies of samples exceeding index limits. The significance of the ingredients “fish” (cat food only) and “liver” (all samples) was explored by calculating the relative risk. The relative risk (syn. risk ratio) is the ratio of the probability of an outcome (e.g., higher amine indexes) in an exposed group (e.g., fish present in canned cat food) to the probability of an outcome in an unexposed group [39]. The relative risk was considered statistically significant when both boundaries of the 95% confidence interval (C.I.) were either below or above 1 [39].

## 3. Results

### 3.1. Information on the Label

Moisture contents, as indicated in the label, ranged from 72% to 90% (average 80.9 ± 2.1), with no statistically significant difference between cat food and dog food. Samples were obtained from 24 different brands (14 for dogs, 13 for cats), with a median of four different products per brand. The declaration of ingredients was in conformity with the requirements of Reg. (EC) No. 767/2009 [40]. Whilst the meat species were mentioned, the type of organs processed into the food was not always stated or clearly stated. According to the label, all samples contained animal byproducts, with liver most often explicitly mentioned (11.3%). Likewise, for some samples, the exact percentages of ingredients were listed, and on others just the minimum amount (i.e., “> 4%”) was stated. In 16.7% of cat food samples, fish was listed as an ingredient.

### 3.2. Amine Contents, All Samples

Contents of putrescine, cadaverine and tyramine were below the limit of detection in most samples (96.8%; 71.5%; 81.7%), with maximum values of 21.5, 98.4, and 32.5 mg/kg wet weight, respectively. Median and maximum contents of histamine, spermidine, and spermine are presented in Table 1. The sum of all six amines was <300 mg/kg in all samples (maximum value: 207.4 mg/kg). Results for index values [27,38] are presented in Table 2. 

### 3.3. Amine Contents, Cat vs. Dog Food

Median amine contents for histamine, spermidine, and spermine were 8.4, 11.5, and 28.2 in cat food, and 25.7, 14.1, and 31.4 mg/kg in dog food (see Table 1). Whilst cat food contained significantly higher amounts of tyramine (i.e., significantly more samples with tyramine contents above the limit of detection), dog food contained significantly higher amounts of histamine and spermine.

No significant differences were found for the frequency of samples with BAI > 1 (*p* = 0.586) [27] and the index according to [38] (*p* = 0.859), see also Table 2.

### 3.4. Significance of Listed Ingredients for Amine Contents 

In canned cat food, the ingredient “fish” was identified as a statistically significant risk factor for BAIs > 1 (relative risk = 3.0 (95% C.I.: 1.8–5.6)) and for contents of (putrescine + cadaverine + histamine + tyramine) exceeding 50 mg/kg (relative risk = 2.4 (95% C.I.: 1.2–4.6)). This could be explained by higher contents of cadaverine in food samples containing fish (median 28.05 mg/kg, whereas the median in samples without fish was below the limit of detection).

Since liver was listed in 21/186 samples, we studied whether the ingredient “liver” would represent a risk factor for higher BAIs or a higher frequency of samples with the sum of (putrescine + cadaverine + histamine + tyramine) exceeding 50 mg/kg. Relative risk was 1.1 (95% C.I.: 0.5–2.2) and 1.1 (95% C.I.: 0.5–2.4), and was thus not statistically significant.

## 4. Discussion

Several researchers have proposed the use of contents of biogenic amines to assess the “freshness” of muscle foods, offal, and seafood [9,28,38]. Since amines are stable to heat processing, amine contents in canned, sterilized foods give an indication on the hygienic condition of the ingredients. This rather straightforward approach is somewhat hampered by the fact that most canned pet food is a mix of various components of animal and non-animal origin [5], and that bacteria or enzymes may remain active in the time from canning to sterilization [41]. Thus, the use of quality indicator values calculated from amine contents indicate that deficiencies in one or more of the ingredients and/or in the processing steps prior to sterilization have occurred. General indicators have been suggested, i.e., the sum of amines not exceeding 300 mg/kg wet weight [26], which was met by all tested samples, and also by samples tested in a previous survey [7]. The proposed freshness limit of a maximum 50 mg/kg [38] was exceeded in 22.6% of samples, which is significantly (*p* = 0.003) less than reported for beef-based canned dog food (44.4%; [7]). This difference persisted when only dog food samples from our study were compared. This was not unexpected, since the median amine contents in dog food samples were, with the exception of spermine, lower than previously reported for canned dog food containing high amounts of beef and byproducts [7]. Given the fact that there are considerable variations between different batches of the same type of raw material [30,32,33], and that we studied only one can from one batch of a product, we refrained from trying to explain this by possible differences in the lists of ingredients.

We observed significantly higher histamine contents in dog food than in cat food, which was not expected when fish would be considered as a primary risk ingredient, since fish was not included in dog food formulations. The median amount of histamine in canned dog food (25.7 mg/kg) matched well with the previously reported 31.6 mg/kg in canned beef-based dog food, but was higher than in beef-based BARF material (16.1 mg/kg; [7]). This might be an indication that canned dog food contains histamine sources other than animal protein or fish; this issue clearly deserves further studies on the amine contents of the ingredients. Although alimentary histamine has been implicated in health disorders in companion animals [17,18,19], it is not easy to establish the toxicological limits; the European Pet Food Industry society [25] sets a 500 mg/kg limit, which is obviously oriented towards the food safety criterion for certain fish species in EU legislation (M = 400 mg/kg; [42]), which is in line with the aim that ingredients for pet food have to be food-grade quality (Austrian Pet Food Union, https://oehtv.at; accessed 10 September 2021). However, this limit was not exceeded in any of our samples.

We could not identify fish as a risk factor for higher histamine contents. However, amongst cat food samples, those containing fish exhibited higher cadaverine contents, which ultimately lead to a significantly higher risk for a BAI > 1 or for the sum of (putrescine + cadaverine + histamine + tyramine) exceeding 50 mg/kg. This higher content of cadaverine is in line with findings that fish spoils easily, with the formation of several biogenic amines but histamine formation being limited to certain fish families only [15]).

In contrast to amines related to amino acid degradation and spoilage, the polyamines spermidine and spermine (and, to some extent, their precursor putrescine) are expected in higher concentrations in tissues active in synthesis and regeneration [29,43]. Since pet food contains slaughter byproducts, this is a source for polyamines in the diet. The usefulness of dietary polyamines in humans is currently under study [24,25], and it will be interesting to observe if a positive effect for companion animals can be established.

Further studies, including the testing of ingredients, intermediates, and finished products along the production chain, are suggested in order to better understand differences in amine patterns between canned foods for dogs and cats, and to explore the role of other factors’ contribution to variations in amine contents (e.g., batch-to-batch variations in the ingredients).

## Figures and Tables

**Table 1 foods-10-02365-t001:** Contents (mg/kg fresh weight) of histamine, tyramine, spermidine, and spermine in canned dog and cat food.

		Histamine		Spermidine		Spermine	
Samples	n	Median ^1^	Max	Median ^1^	Max	Median ^1^	Max
All	186	14.5 (LOD-29.8)	61.7	12.7 (9.21–15.3)	28.2	29.4 (20.5–35.7)	53.6
For dogs	72	25.7 ^a^ (14.2-32.6)	52.2	14.1 (11.7–17.2)	28.2	31.4 ^b^ (27.3–38.8)	53.6
For cats,							
All	114	8.4 ^a^ (LOD-22.2)	61.6	11.5 (8.1–14.7)	22.0	28.2 ^b^ (17.6–33.1)	52.9
With fish	19	8.8 (LOD-16.1)	27.4	8.9 (7.7–11.3)	13.6	19.3 (16.0–27.2)	31.3
Without fish	95	8.3 (LOD-22.8)	61.6	12.0 (8.5–15.2)	22.0	29.2 (18.1–34.6)	52.9

Note: 1 = First and third quartiles in brackets; LOD = below limit of detection; Max = maximum value; within columns, data with the same superscript letters indicate statistically significant differences between dog and cat food, *p* < 0.05.

**Table 2 foods-10-02365-t002:** Results for amine index values in canned dog and cat food.

			BAI [27]		Sum of Putrescine, Cadaverine, Histamine and Tyramine [38]	
Samples		n	Median (Maximum)	% of Samples with BAI > 1	Median (Maximum)	% with Sum > 50 mg/kg
All		186	0.62 (2.65)	26.7	31.1 (156.8)	22.6
Dog only		72	0.68 (2.65)	25.0	34.7 (156.8)	23.6
Cat only	all	114	0.51 (2.46)	28.1	24.4 (141.1)	21.9
	with fish	19	1.19 (2.29)	63.2 ^a^	47.2 (129.8)	42.1^b^
	without fish	95	0.45 (2.46)	21.1 ^a^	18.4 (141.1)	17.9^b^

Note: Within columns, data with the same superscript letters differ statistically significantly, *p* < 0.05.

## Data Availability

The dataset analyzed in this study is publicly available. This data can be found here: https://phaidra.vetmeduni.ac.at/o:812 (accessed 30 September 2021).

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
