# Peer review of "Contents of Polyamines and Biogenic Amines in Canned Pet (Dogs and Cats) Food on the Austrian Market"

_foods, 2021, doi:10.3390/foods10102365_

Round 1

Reviewer 1 Report

REVIEW of Foods-140897

The manuscript deals with the occurrence of biogenic amines in pet foods, distinguishing the samples based on the ingredients (cat food, with fish; dog food, without fish)

The topic is relevant

I suggest doing some revisions in order to better enhance the data obtained.

In general, some sections should be revised in accordance with the instructions for the authors (for example, at line 196: please delete Paulsen et al. 2021; please check it in all paper).

In my opinion, in all the text fractions it could generate confusion if they are repeatedly reported in the text; wouldn't it be better to report a percentage?

Authors should better specify what they means by relative risk RR. This is an important point to be clarified in the paper.

Abstract:

Results could be better reported as (min-max; median ± st.dev.);

line 19: please add "respectively" at the end of sentence.

lines 24-26: in general abbreviation in abstract should be avoided

Introduction

line 86: please add "(BAI)"

line 95: what is BARF? Please authors to explain the extended meaning of the acronym BARF, as well as of the other acronyms the first time they mention them in the text.

Materials and Methods

I suggest separating this section in more subsections (i.e. sampling, chemical analysis, statistical treatment of data). I suggest also a brief subsection in which two formulas used and reference limits were reported

i.e.       BAI =                      limits                       (references)

Moreover authors could specify the mean of relative risk (RR) and C.I. in this section

Results

Lines 144-176: data could be shown in a big table reporting: % sample, range (min-max), median ± st.dev.; BAI index, sum of BA, statistical significant test results. A Table for two class of pet foods could be add. In this way results are more understandable.

Figure 1: please add the title of "x" axis. Then, The caption of figure could be "Amine contents (mg/ Kg fresh matter)...."

Discussion

How authors could correlate high histamine level in dog food?

 Data could be better discussed

line 190: Midwest? what does it mean? Please explain

line 212: "are" or "have" to be food-grade...?

line 213: "However, this limit was not..."

lines 215-220. This paragraph is unclear

Author Response

Dear reviewer, thank you for your comments which we addressed in the revision of the manuscript as described below. Answers are in italics.

In general, some sections should be revised in accordance with the instructions for the authors (for example, at line 196: please delete Paulsen et al. 2021; please check it in all paper).

We have removed authors´names in the text and assured that the references are quoted correctly by numbers in square brackets.

In my opinion, in all the text fractions it could generate confusion if they are repeatedly reported in the text; wouldn't it be better to report a percentage?

            Fractions have the advantage that they also report the umber of samples. We agree that in this case, with higher numbers of samples, percentages may aslo be fine and changed this accordingly.

Authors should better specify what they means by relative risk RR. This is an important point to be clarified in the paper.

„Relative risk“ (syn. risk ratio, RR) is the ratio of the probability of an outcome in an exposed group to the probability of an outcome in an unexposed group.In this case „exposed“ means cat foods with fish as ingredient, and the outcome would be BAI >1 or max. 1. We inserted an explanation in the materials and methods section.

Abstract:

Results could be better reported as (min-max; median ± st.dev.);

            Since results for most amines were not normally distributed or a substantial fraction was below limit of detection, we decided to add maximum values to the abstract and to keep median instead of mean values, and to add quartiles.

line 19: please add "respectively" at the end of sentence.

            We added „respectively“

lines 24-26: in general abbreviation in abstract should be avoided

We have now written out all abbreviations.

Introduction

line 86: please add "(BAI)"

            Added.

line 95: what is BARF? Please authors to explain the extended meaning of the acronym BARF, as well as of the other acronyms the first time they mention them in the text.

            We have now included explanations of the abbreviations at first mention.

Materials and Methods

I suggest separating this section in more subsections (i.e. sampling, chemical analysis, statistical treatment of data). I suggest also a brief subsection in which two formulas used and reference limits were reported

i.e.       BAI =                      limits                       (references)

Moreover authors could specify the mean of relative risk (RR) and C.I. in this section

            We have now structured the material and methods section accordingly.

Results

Lines 144-176: data could be shown in a big table reporting: % sample, range (min-max), median ± st.dev.; BAI index, sum of BA, statistical significant test results. A Table for two class of pet foods could be add. In this way results are more understandable.

            Thank you very much for this suggestion. We have now put the information in 2 tables and removed Figure 1, in order to avoid duplicate data reporting.

Figure 1: please add the title of "x" axis. Then, the caption of figure could be "Amine contents (mg/ Kg fresh matter)...."

            We have adopted the titles of the x-axis and the caption of the figure. According to reviewers suggestions, we reoved the figure and present the information now in a comprehensive Table.

Discussion

How authors could correlate high histamine level in dog food? Data could be better discussed

            We have currently no explanation for that, but since we observed a similar issue in a previous study, we will conduct further studies on the amine contents of ingredients. This is expressed more clearly now in the revised manuscript.

line 190: Midwest? what does it mean? Please explain

This was a mistake – it means reference no. 26. Now corrected.

line 212: "are" or "have" to be food-grade...?

            Thank you, now corrected: .. have to be food grade quality“

line 213: "However, this limit was not..."

            Corrected.

lines 215-220. This paragraph is unclear

            We have corrected „amines“ to „cadaverine“ and inserted some words for clarification: We could not identify fish as a risk factor for higher histamine contents. However, amongst cat food samples, those containing fish exhibited higher cadaverine contents, which ultimately lead to a significantly higher risk for a BAI >1 or for the sum of (putrescine+cadaverine+histamine+tyramine) exceeding 50 mg/kg. This higher content of cadaverine is in line with findings that fish spoils easily, with formation of several biogenic amines but histamine formation being limited to certain fish families only [15]).

Reviewer 2 Report

The article summarizes a study about contents of biogenic amines and polyamines as quality indicators of canned dog and cat foods sampled in Austria. The manuscript needs minor revision.

Abstract

Line 12, I would leave out “will” in the first sentence.

Introduction

Line 70, either due to the abundance of

Line 92, … dogs and cats. In pet food industry, there …

Materials and methods

Line 115, to which pH was the sample titrated with NaHCO3?

Results

Line 150, please add a reference describing how the BAI is calculated

Discussion

Line 191, was exceeded

Line 206, food

Author Response

Dear reviewer, thank you for your comments which we addressed in the revision of  the manuscript as described below. Answers are in italics.

Abstract

Line 12, I would leave out “will” in the first sentence.

            Thank you, we removed the word „will“.

Introduction

Line 70, either due to the abundance of

            Thank you, corrected.

Line 92, … dogs and cats. In pet food industry, there …

                        Thank you, corrected.

Materials and methods

Line 115, to which pH was the sample titrated with NaHCO3? pH ca. 11

Results

Line 150, please add a reference describing how the BAI is calculated

added

Discussion

Line 191, was exceeded

                        Thank you, corrected.

Line 206, food

                        Thank you, corrected.